# Multidrug-Resistant Enterobacterales in Community-Acquired Urinary Tract Infections in Djibouti, Republic of Djibouti

**DOI:** 10.3390/antibiotics11121740

**Published:** 2022-12-02

**Authors:** Hasna Said Mohamed, Mohamed Houmed Aboubaker, Yann Dumont, Marie-Noëlle Didelot, Anne-Laure Michon, Lokman Galal, Hélène Jean-Pierre, Sylvain Godreuil

**Affiliations:** 1Laboratoire de Bactériologie, Centre Hospitalier Universitaire de Montpellier, 34295 Montpellier, France; 2MIVEGEC, IRD, CNRS, Université de Montpellier, 34394 Montpellier, France; 3Hospital General Peltier de Djibouti, Djibouti City 2123, Djibouti; 4Laboratoire de Biologie Médicale de la Mer Rouge, Djibouti City 1119, Djibouti; 5Laboratoire de la Caisse Nationale de Sécurité Sociale, Djibouti City 696, Djibouti; 6Jeune Equipe Associée à IRD (JEAI), FASORAM, 34394 Montpellier, France

**Keywords:** urinary tract infections, Enterobacterales, extended-spectrum beta-lactamase carbapenemases, community, Djibouti

## Abstract

The emergence and spread of multidrug resistant Enterobacterales (MDR-E) are a global public health issue. This problem also concerns urinary tract infections (UTI), which are the second most frequent infections after respiratory infections. The objective of this study was to determine MDR-E frequency and to characterize MDR-E isolates from patients with community-acquired UTIs in Djibouti, Republic of Djibouti. From 800 clinical urinary samples collected at the Mer Rouge Laboratory, Djibouti, from January to July 2019, 142 were identified as Enterobacterales (age range of the 142 patients mean age is 42 years.) Mass spectrometry analysis of these isolates identified 117 *Escherichia coli*, 14 *Klebsiella pneumoniae*, 2 *Proteus mirabilis,* 4 *Enterobacter* spp., 4 *Providencia stuartii* and 1 *Franconibacter helveticus*. Antibiotic susceptibility testing (disk diffusion method) of these 142 isolates detected 68 MDR-E (68/142 = 48%): 65 extended-spectrum bêta lactamase- (ESBL), 2 carbapenemase- (one also ESBL), and 1 cephalosporinase-producer. Multiplex PCR and sequencing showed that the 65 ESBL-producing isolates carried genes encoding CTX-M enzymes (CTX-M-15 in 97% and CTX-M-9 in 3% of isolates). Two isolates harboured a gene encoding the OXA-48-like carbapenemase, and one the gene encoding the AmpC CMY-2 cephalosporinase. Genes implicated in resistance to quinolones (qnrB, aac (6′)-Ib-cr, qnrD, oqxA and B) also were detected. Among the *E. coli* phylogroups, B2 was the most common phylogenetic group (21% of MDR-E isolates and 26% of non-MDR-E isolates), followed by A (14% and 12%), B1 (9% and 7%), D (3% and 3%), F (3% and 3%) and E (2% and 2%). This study highlights the high frequency of ESBL producers and the emergence of carbapenemase-producers among Enterobacterales causing community-acquired UTIs in Djibouti.

## 1. Introduction

Community-acquired urinary tract infections (UTI) are a public health problem worldwide and the second most frequent infections after respiratory infections [1].

Importantly, UTI epidemiology has changed due to the emergence of extended spectrum beta-lactamase (ESBL)-producing Enterobacterales (ESBL-E) [2,3]. Until the late 1990s, the majority of identified ESBLs were TEM-1/2 or SHV-1 mutants. The producing strains were often associated with nosocomial epidemics, and the prevalence of ESBL producers was higher in *Klebsiella pneumoniae* than in *Escherichia coli* isolates [4,5].

However, in the late 1980s–early 1990s, new ESBL types were reported in Europe (MEN-1, CTX-M-1, PER-1), Japan (SFO-1, Toho-1), Argentina (CTX-M-2, PER-2) and Mexico (TLA-1) [4,6,7]. Since the 2000s, CTX-M-producing *E. coli* strains are the main ESBL-E detected in adult and paediatric UTIs, including in community settings. This phenomenon has accelerated in recent years, and CTX-M is now the main ESBL worldwide. The significant increase in ESBL-E prevalence has led to a parallel increase in carbapenem prescriptions in hospital and community settings [3,8,9]. Consequently, recently, Enterobacterales producing beta-lactamases that hydrolyse class A, B and D carbapenems have been identified worldwide [10]. OXA-48-type (class D) carbapenemases were first detected in *K. pneumoniae* in Turkey in 2001 [11]. Since then, Enterobacterales producing these enzymes have been found worldwide [12,13]. This is a public health issue because these bacteria are often ESBL-producers, and may also exhibit resistance to other antibiotic classes, particularly fluoroquinolones and aminoglycosides [14]. Therefore, the objective of this study was to determine the rate of multidrug-resistant Enterobacterales that produce ESBL, AmpC beta-lactamases, carbapenemases and beta-lactamases, as well as their antibiotic resistance profiles in patients with UTI in Djibouti, Republic of Djibouti.

## 2. Results

### 2.1. Bacterial Isolates and Patient Characteristics

Among the 327 bacterial strains isolated from patients with community-acquired UTIs, 200 were Enterobacterales (61.2%), 93 were Gram-positive cocci (28.4%; *Enterococcus* spp., *Streptococcus spp.* and *Staphylococcus* spp.), and 34 were Gram-negative non-fermenting bacilli (10.4%, *Acinetobacter* spp. and *Pseudomonas aeruginosa*). However, only 142 (71%) of the 200 Enterobacterales isolates could be analysed. Enterobacterales were represented by seven species: *E. coli* (*n* = 117, 82.4%), *K. pneumoniae* (*n* = 14, 9.8%), *Providencia stuartii* (*n* = 4, 2.8%), *Enterobacter cloacae* (*n* = 3, 2.2%), *Proteus mirabilis* (*n* = 2, 1.4%), *Enterobacter kobei* (*n* = 1, 0.7%) and *Franconibacter helveticus* (*n* = 1, 0.7%). The age of these 142 patients ranged from 1 to 85 years, and 61 (43%) were men.

### 2.2. Multidrug-Resistant Enterobacterales (MDR-E) Isolates

Antimicrobial susceptivity testing indicated that among these 142 isolates, 68 (48%) were MDR-E and 77 were non-MDR-E (53.5%). PCR analysis indicated that 65/68 (95.5%) produced ESBL (*n* = 58, 41%, *E. coli* and *n* = 7, 5%, *K. pneumoniae*), 1 *E. coli* isolate (1%) produced ESBL and carbapenemase, and 1 *E. coli* isolate (1%) 1 cephalosporinase. Analysis of the risk factors showed no significant difference in the MDR-E and non-MDR-E percentages in the function of the patients’ age (Table 1). Conversely, the MDR-E rate was significantly higher in women than men (43% vs. 57%, *p* < 0.005), and in patients who reported previous antibiotic use (53% vs. 18.4%, *p* < 0.005).

### 2.3. Antibiotic Resistance Patterns

Among the 142 Enterobacterales isolates, the percentage of antibiotic-resistant isolates was higher in the MDR-E than non-MDR-E group: to fluoroquinolone (86.3% vs. 22.9%; *p* < 0.005), aminoglycosides (56% vs. 5.3%; *p* < 0.005), tetracycline (77.3 vs. 32.9%; *p* < 0.005) and sulfamethoxazole-trimethoprim (77.3 % vs. 30.3%; *p* < 0.005) (Figure 1). In both groups, few isolates were resistant to fosfomycin (15.1% and 10.5%) and chloramphenicol (87.46% and 12.49%). All 142 isolates were sensitive to colistin (MIC < 2 mg/L). Two carbapenemase-producing isolates were resistant to ertapenem.

### 2.4. Molecular Characterization of Beta-Lactamases and Encoding Genes

The results of the PCR and sequencing analyses showed that CTX-M group 1 was the most frequent ESBL type (97% of isolates; 96% of E. coli and 100% of K. pneumoniae isolates) and was encoded exclusively by the blaCTX-M-15 gene (Table 2). Two E. coli isolates harboured the CTX-M group 9 ESBL type and the blaCTX-M-14 gene. Genes encoding CTX-M ESBL were detected alone in three samples (4.6%), and associated with one or two other beta-lactamase-encoding genes (blaTEM and blaOXA) in 62 isolates (95.4%). The blaSHV gene was not detected. Two CTX-M-15-producing E. coli isolates co-expressed the OXA-48 enzyme, and one non-ESBL-producing E. coli isolate harboured the gene encoding the AmpC enzyme CMY-2 (Table 2). Genes involved in resistance to quinolones, such as qnrB (*n* = 2 MDR-E isolates), aac (6′)-Ib-cr (*n* = 15 MDR-E isolates), qnrD (*n* = 3 MDR-E isolates) and oqxA and B (*n* = 5 MDR-E isolates) also were detected (Table 2). The armA, rmtA and rmtB genes, implicated in resistance to aminoglycosides, were not detected.

### 2.5. Molecular Epidemiology Typing

The phylogenetic group assignment of the 117 *E. coli* isolates (*n* = 59 MDR-E and *n* = 58 non-MDR-E) is summarized in Table 3. B2 was the most common phylogenetic group (21% of MDR-E isolates and 26% of non-MDR-E isolates), followed by A (14% and 12%), B1 (9% and 7%), D (3% and 3%), F (3% and 3%) and E (2% and 2%).

The sequence type (ST) for MDR-E *E. coli* isolates and MDR-E *K. pneumoniae* isolates is shown in Table 2.

## 3. Discussion

Over the last decade, Gram-negative ESBL-E have emerged as serious pathogens in hospitals and in the community worldwide [15]. As the emergence of ESBL-E varies in different regions of the world [16], antibiotic resistance must be precisely monitored to propose empirical treatment policies. The increase of ESBL-E is a heavy burden for the management of community-acquired UTIs because these isolates are often MDR-E, increasing the risk of treatment failure [16,17]. In our study, 68 (48%) Enterobacterales isolates were MDR-E. This is lower than in other studies [18,19,20], where the MDR-E rate ranged from 50% to 68.0% and 74.2%. ESBL production was detected in 65 (46%) MDR-E isolates: 58 (41%) *E. coli* and 7 (5%) *K. pneumoniae* isolates. In previous studies, the ESBL producer rate among isolates from patients with UTI (mainly community-acquired) was variable: 79% in Lebanon [21], 17% in Egypt [22], 6.7% in Libya [15,23] and 0–2.4% in Spain [19].

Among the different ESBLs, CTX-M enzymes are the most frequently detected in different epidemiological settings. The number of reports on community-acquired CTX-M-producing *E. coli* strains (disease-causing or colonizers) is on the increase [24]. In this study, the predominant ESBL-encoding genes were blaCTX-M (97% of isolates) and blaTEM (3% of isolates). This is in agreement with other studies showing that the CTX-M type has replaced the TEM and SHV types in Enterobacterales isolates in several countries, both in nosocomial and community settings [25,26]. For instance, the CTX-M type was predominant (71.4%) in isolates from urine in Egypt (urinary and stool samples) [22,27]. The relatively higher frequency of blaCTX-M in community-acquired isolates in our study is consistent with CTX-M ESBLs originating from environmental bacteria, unlike TEM or SHV ESBLs [28]. Among the blaCTX-M genes, blaCTX-M-15 was the most frequent (95.4%) in our study, as previously reported, thus highlighting its global spread by clonally related *E. coli* strains [15].

Carbapenems are considered one of the last-line treatments available to treat infections caused by MDR Gram-negative bacteria. Therefore, the emergence of carbapenemase-producing Enterobacterales represents a major public health problem [29]. The first reported OXA-48-producing Enterobacterales isolate was a *K. pneumoniae* strain isolated in Turkey in 2001 as the cause of hospital-acquired infections [11]. OXA-48-like enzymes have been found worldwide in Enterobacterales isolates and have been widely reported as the source of several hospital outbreaks. Recently, the blaOXA-48 gene has been detected in community isolates in many countries [29,30]. Here, we identified the first two *E. coli* isolates co-producing β-lactamases OXA-48 alone or with CTX-M15 in Djibouti. The first Algerian community-acquired UTI caused by a *K. pneumoniae* isolate that co-produced the β-lactamases OXA-48 and SHV-27 was described in 2017 [31]. In addition, one *E. coli* isolate produced the CMY-2 enzyme, which is the most common AmpC in *E. coli* [32]. The low frequency of AmpC in UTI-causing *E. coli* isolates has been reported also in Algeria [33].

This study revealed a relatively high resistance rate to most of the antibiotics tested. Resistance to trimethoprim, sulfamethoxazole/amoxicillin/clavulanic acid and ampicillin was particularly alarming. This is consistent with findings in Cameroun [34] and Uganda [35]. This may be explained by the easy access to these antibiotics and their massive and uncontrolled use in Djibouti. The particularly high resistance rates of our isolates to trimethoprim/sulfamethoxazole (cotrimoxazole) could be explained also by other factors specific to our setting, such as the use of cotrimoxazole for prophylaxis in HIV-infected patients and the use of the sulfadoxine/pyrimethamine combination, which shares enzymatic targets with cotrimoxazole, for routine malaria prophylaxis during pregnancy, as previously suggested by a study carried out in Tunisia [36]. As 73.8% of our isolates were sensitive to temocillin, this antibiotic could be a good alternative for the management of UTIs caused by ESBL-E.

In our study, 50.34% of Enterobacterales isolates were resistant to ciprofloxacin and 55.17% to ofloxacin. According to Guessennd et al. [37], three types of genes are involved in the resistance of Enterobacterales to quinolones: the “qnr” genes, the genes encoding N-acetyl transferase and the genes encoding the QepA efflux pump. The association of these genes with β-lactam and aminoglycoside resistance mechanisms can induce resistance to fluoroquinolones (e.g., ciprofloxacin, ofloxacin). Our resistance rates to ciprofloxacin and ofloxacin were higher than those reported in other studies (40% for ciprofloxacin and 9.1% for ofloxacin in Cameroon [34], 26% for ciprofloxacin and 35% for ofloxacin in Gabon [38] and 13.75% for ciprofloxacin and 17.5% for ofloxacin in Egypt [39]. This difference could be explained by the first-line use of fluoroquinolones as probabilistic treatment of community-acquired UTIs in Djibouti [40]. Amikacin (11.27%) was the most active antibiotic among aminoglycosides. Aminoglycoside resistance was similar to that reported in Iran (16.7% and 21.8%) [41] and Morocco (8% and 14%) [42]. The apparently preserved efficacy of aminoglycosides could be explained by their frequent parenteral administration, which limits their use. Imipenem and fosfomycin also have maintained excellent activity and could be adopted as therapeutic alternatives. These results are in agreement with Tunisian [36] and European data [43].

Phylogenetic analysis is important to identify new groups of emerging bacteria. Most isolates were in the B2 group (43% of isolates), followed by the A (26%), B1 (15%), D (6%), F (6%) and E (4%) groups. In other studies [36,44,45] also, the B2 subgroup was the most common group, especially among CTX-M15-producing *E. coli* strains. Several studies have shown that CTX-M-15-producing *E. coli* are among the most prevalent ESBL-producing Enterobacterales species [46] and that the worldwide dissemination of ESBL-producing *E. coli* is associated with specific clones that harbour a plasmid carrying the blaCTM-X-15 gene. In some African countries, CTX-M-15-producing *E. coli* belonging to the phylogenetic groups A and D have been detected in extra-intestinal infections [22,47,48,49,50,51].

## 4. Materials and Methods

### 4.1. Study Setting

All consecutive urinary samples from non-hospitalized patients with UTI sent to the Mer Rouge Medical Biology Laboratory in Djibouti from January to July 2019, were included in this study. The Mer Rouge Medical Biology Laboratory is the main private medical microbiology analysis laboratory in Djibouti, the capital city, with a population of ~620,000 inhabitants. In 2019, this laboratory received more than 1560 biological samples, including 1380 (88.5%) urine samples for microbiology analysis. This study was approved by the Djiboutian Ministry of Public Health ethics board (No 104/IGSS/2017/MS).

### 4.2. Specimen Collection, Identification, and Antimicrobial Susceptibility Testing

During the study period, 800 urinary samples were sent to the laboratory for bacteriologic investigations. From these urinary specimens, 327 (41%) non-duplicated and clinically significant bacterial isolates were from outpatients with UTI. Bacterial species were identified by matrix-assisted laser desorption ionization–time of flight mass spectrometry (Bruker Daltonics, Bremen, Germany). Antimicrobial susceptibility was tested with the disk diffusion method on Müller-Hinton agar. The following antibiotics were tested: amoxicillin, amoxicillin-clavulanic acid, aztreonam, cefepime, cefotaxime, cefpirome, cefpodoxime, cefoxitin, ceftazidime, cephalothin, moxalactam, piperacillin, piperacillin-tazobactam, ticarcillin, ticarcillin-clavulanic acid, imipenem, nalidixic acid, ciprofloxacin, levofloxacin, ofloxacin, amikacin, gentamicin, netilmicin, tobramycin, fosfomycin, chloramphenicol, tetracycline and trimethoprim-sulfamethoxazole. Results were interpreted according to the European Committee on Antimicrobial Susceptibility Testing (EUCAST) guidelines and clinical breakpoints (http://www.eucast.org/clinical_breakpoints/ accessed on 27 November 2022). ESBL production was detected with the combined double-disk synergy method. In the case of elevated cephalosporinase production, the combined double-disk synergy test was performed using cloxacillin-supplemented medium. Ertapenem minimal inhibitory concentrations (MICs) were determined with the Etest method (bioMérieux, Marcy-l’Etoile, France) carbapenemase-producing *E. coli. Colistin* MIC was determined in liquid medium using the UMIC *Colistin*^®^ kit (Biocentric, Bandol, France).

### 4.3. Molecular Identification of ESBLs and Carbapenemases

DNA was extracted from one single colony of each isolate and incubated in 100 mL of distilled water at 95 °C for 10 min, followed by centrifugation. The presence of the blaNDM, blaOXA48-like, blaGIM, blaPER, blaIMP, blaVIM, blaSPM, blaKPC, blaDIM, blaSIM, blaBIC, blaAIM, blaVEB, blaCTX-M (CTX-M group 1, 2, 8, 9 and 25), blaTEM, blaSHV and blaOXA-1-like genes; AmpC-type-producing genes (blaADC, blaFOX, blaMOX, blaDHA, blaCMY and blaMIR); the aminoglycoside resistance-conferring 16S rRNA methylase genes (armA, rmtA, rmtB, rmtC and rmtD); and the plasmid-mediated quinolone resistance (PMQR) genes (qnrA, qnrB, qnrS, qnrC, qnrD, qepA, aac(6′)-Ib-cr and oqxA and B) was assessed by multiplex PCR using a previously published method [12,13]. DNA samples from reference ESBL-, carbapenemase-, 16S rRNA methylase- and PMQR-positive strains [52,53] were used as positive controls. PCR products were visualized by electrophoresis on 1.5% ethidium bromide-containing agarose gels at 100 V for 90 min. A 100 bp DNA ladder (Promega) was used as marker size. PCR products were purified using the ExoSAP-IT PCR Product Clean-up Reagent (GE Healthcare, Piscataway, NJ, USA), and sequenced bidirectionally on a 3100 ABI Prism Genetic Analyzer (Applied Biosystems). Nucleotide sequence alignment and analyses were performed online using the BLAST program available at the National Center for Biotechnology Information (www.ncbi.nlm.nih.gov accessed on 27 November 2022).

### 4.4. Molecular Epidemiology Typing

To determine the phylogenetic group of *E. coli* isolates, the new PCR-based method described by Clermont et al. [54] was used. This method uses modified primers for chuA, yjaA and TspE4.C2 that eliminate some primer mismatches, and, importantly, allows distinguishing strains belonging to the phylogroups C, E, F and clade I. Multilocus sequence typing (MLST) was performed as described in the Institut Pasteur MLST database (http://bigsdb.pasteur.fr (accessed on 27 August 2019) for *E. coli* and *K. pneumoniae*.

### 4.5. Statistical Analysis

Statistical analyses were performed with the R software. Different variables were compared with the Chi-square (χ^2^) test. Differences were considered statistically significant at the 0.05 confidence level.

## 5. Conclusions

Our study demonstrated that ESBLs, CTX-M type and phylogenetic group B2 are prevalent in community-acquired UTI-causing *E. coli* isolates in Djibouti. We identified the first ESBL-E isolates that co-produced CTX-M, OXA-1 and TEM-1 enzymes in Djibouti. The emergence of strains harbouring several beta-lactamases simultaneously is a major problem. More studies are needed to monitor the spread of MDR-E in the Djiboutian population.

## Figures and Tables

**Figure 1 antibiotics-11-01740-f001:**
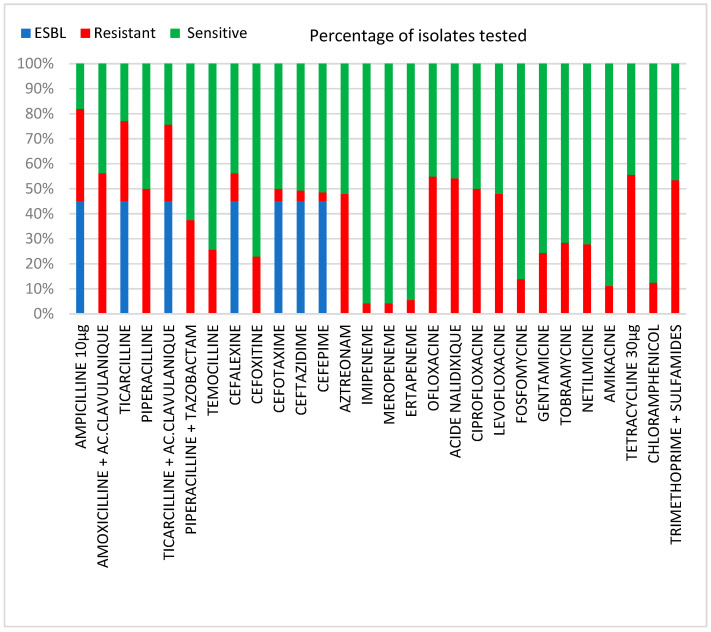
Antibiotic resistance profile of Enterobacterales isolates.

**Table 1 antibiotics-11-01740-t001:** Characteristics of urinary Enterobacterales studied according to their production of ESBL or not.

Factors.	ESBL-Producing (*n* = 65)	non-ESBL-Producing (*n* = 77)
Age (years)	N (%)	N (%)
15–24 years	19 (13%)	30 (21%)
25–49 years	32 (23%)	24 (17%)
≥50 years	14 (10%)	23 (16%)
Sex		
Male	21 (15%)	40 (28%)
Female	44 (31%)	37 (26%)
Previous use of antibiotics (last 3 months)	35 (54%)	14 (18%)

**Table 2 antibiotics-11-01740-t002:** Molecular characterization of *Escherichia coli* and *Klebsiella pneumoniae* isolates from outpatients with urinary infection.

Isolate	Beta-Lactamases	Carbapenemases	Phylogroup	Sequence Type	Resistance to Quinolones
*E. coli*13	CTX-M-15+OXA-1		B2	ST-43	qnrB
*E. coli*7U		OXA48	A	ST2450	
*E. coli*49	CTX-M-15+OXA-1		B2	ST-43	qnrA
*E. coli*53	CTX-M-15+OXA-1		B2	ST-43	
*E. coli*48	CTX-M-15+OXA-1		B2	ST-43	qnrA
*E. coli*56	CTX-M-15+TEM-1, OXA-1		B2	ST-43	
*E. coli*95	CTX-M-15+OXA-1		B2	ST-43	
*E. coli*64	CTX-M-15+TEM-1, OXA-1		B2	ST-43	qnrD
*E. coli*11	CTX-M-15+TEM-1, OXA-1		B2	ST-43	qnrC
*E. coli*24	CTX-M-15+OXA-1		B2	ST-43	qnrC
*E. coli*43	CTX-M-15+OXA-1		B2	ST53	qnrD
*E. coli*55	CTX-M-15+OXA-1		F	ST2	
*E. coli*41	CTX-M-15+TEM-1, OXA-1		B2	ST53	oqxAB
*E. coli*49	CTX-M-14+OXA-1		D	ST829	oqxAB
*E. coli*58	CTX-M-15+OXA-1		A	ST692	qnrC
*E. coli*95	CTX-M-15+TEM-1, OXA-1		A	ST698	qnrA
*E. coli*02	CTX-M-14+OXA-1		A	ST2	oqxAB
*E. coli*13	CTX-M-15+TEM-1, OXA-1		B1	ST954	qnrD
*E. coli*75	CTX-M-15+OXA-1		A	ST690	oqxAB
*E. coli*98	CTX-M-15+OXA-1		D	ST44	
*E. coli*91	CTX-M-15		B2	ST-43	
*E. coli*08	CTX-M-15+TEM-1		D	ST44	qnrA
*E. coli*39	CTX-M-15+OXA-1		B2	ST53	qnrA
*E. coli*08	CTX-M-15+TEM-1, OXA-1		B2	ST53	
*E. coli*64	CTX-M-15+OXA-1		B2	ST53	
*E. coli*19	CTX-M-15+TEM-1, OXA-1		B2	ST53	qnrA
*E. coli*14	CTX-M-15+TEM-1		F	ST2	
*E. coli*78	CTX-M-15+OXA-1		F	ST2	
*E. coli*49	CTX-M-15		B2	ST-43	
*E. coli*28	CTX-M-15+OXA-1		B2	ST-43	qnrA
*E. coli*21	CTX-M-15+OXA-1		B2	ST-43	qnrA
*E. coli*37	CTX-M-15+OXA-1		B2	ST-43	
*E. coli*28	CTX-M-15+OXA-1		B2	ST-43	
*E. coli*05	CTX-M-15+OXA-1		B2	ST-43	qnrA
*E. coli*81	CTX-M-15+OXA-1		B2	ST-43	
*E. coli*88	CTX-M-15+TEM-1, OXA-1		B2	ST-43	
*E. coli*11	CTX-M-15+OXA-1		B2	ST-43	
*E. coli*67	CTX-M-15+TEM-1, OXA-1		B2	ST-43	qnrA
*E. coli*32	CTX-M-15+OXA-1		A	ST500	
*E. coli*30	CTX-M-15		B1	ST741	
*E. coli*31	CTX-M-15+OXA-1		B1	ST741	
*E. coli*78	CTX-M-15+OXA-1		B1	ST960	
*E. coli*53	CTX-M-15+TEM-1, OXA-1		B1	ST960	
*E. coli*95	CTX-M-15+OXA-1		B1	ST960	
*E. coli*4U	CTX-M-15+OXA-1	OXA48	C	ST410	
*E. coli*10	CTX-M-15+OXA-1		A	ST692	
*E. coli*54	CTX-M-15+TEM-1		A	ST692	
*E. coli*06	CTX-M-15+OXA-1		A	ST692	
*E. coli*90	CTX-M-15+OXA-1		A	ST692	
*E. coli*30	CTX-M-15+OXA-1		A	ST692	
*E. coli*76	CTX-M-15+TEM-1, OXA-1		A	ST692	
*E. coli* 29	CTX-M-15+OXA-1		A	ST692	
*E. coli*75	CTX-M-15+OXA-1		A	ST692	
*E. coli*90	CTX-M-15+TEM-1		D	ST829	
*E. coli*26	CTX-M-15+TEM-1		D	ST829	
*E. coli*063	CTX-M-15+TEM-1		E	ST244	
*E. coli*63	CTX-M-15+OXA-1		E	ST244	
*E. coli*076	CTX-M-15+TEM-1		F	ST2	
*K.* *pneumoniae*20	CTX-M-15+ OXA-1			ST592	
*K.* *pneumoniae*54	CTX-M-15+ OXA-1			ST464	
*K.* *pneumoniae*03	CTX-M-15+TEM-1			ST29	
*K.* *pneumoniae*11	CTX-M-15+ OXA-1			ST889	
*K.* *pneumoniae*26				ST732	qnrA
*K.* *pneumoniae*43				ST16	
*K.* *pneumoniae*13	CTX-M-15+ OXA-1			ST485	
*K.* *pneumoniae*710	CTX-M-15+ OXA-1			ST889	
*K.* *pneumoniae*50	CTX-M-15+ OXA-1			ST464	

**Table 3 antibiotics-11-01740-t003:** Molecular characterization and Phylogenetic Group Assignment of ESBL-EC.

Phylogenetic Group	MDR-E N (%)	non-MDR-E N (%)	Total ESBL-EC N (%)
B2	24 (21%)	26 (22%)	50 (43%)
A	16 (14%)	14 (12%)	30 (26%)
B1	10 (9%)	8 (7%)	18 (15%)
D	4 (3%)	4 (3%)	8 (6%)
F	3 (3%)	4 (3%)	7 (6%)
E	2 (2%)	2 (2%)	4 (4%)
Total	59 (5%)	58 (50%)	117 (100%)

## Data Availability

Not applicable.

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
