# Peer review of "Multidrug-Resistant Enterobacterales in Community-Acquired Urinary Tract Infections in Djibouti, Republic of Djibouti"

_antibiotics, 2022, doi:10.3390/antibiotics11121740_

Round 1

Reviewer 1 Report

The manuscript entitled "Multidrug-resistant Enterobacteriaceae in community-acquired urinary tract infections in Djibouti, Republic of Djibouti" represents considerable work. The following comments must be addressed before the manuscript is suitable for publication in the antibiotics Journal.

- There are linguistic imprecision and errors in grammar and style leading to misunderstanding; please correct them.

- In abstract: The name of microorganisms and genes must be italic and also throughout the manuscript.

-Line 73 and 74: Please change 1 to one.

-           

Author Response

Some linguistic inaccuracies and grammatical and stylistic errors have led to misunderstandings; please correct them.
 ok
- In summary: Names of microorganisms and genes should be italicized and evenly spaced throughout the manuscript.
ok
-Line 73 and 74: Please replace 1 with 1.

Reviewer 2 Report

In the manuscript submitted by Mohamed et al., the authors report the prevalence of MDR-E as well as the genotypic and phenotypic characterization of several of the MDR-E isolates from patients with UTIs. Overall, the content of the submitted manuscript expands our understanding of the extent to which AMR and MDR bacteria are causing infections worldwide. Although important, there are major concerns with the manuscript as submitted.

Major Concerns:
1. Proofreading is a must! While the language is acceptable, there are numerous errors with citation format or the inclusion of the actual reference.
     a. Please verify throughout the manuscript text which citation format you want to use. It's either text [1] or text[1] with or without the space between the text and the bracket. Select one and stick with it.
     b. Also verify proper spacing at the end of sentences that have citations. See Lines 37, 42, 44, 49, 52, 53, 129, 159, 188, 237, 247.
     c. Line 141 is missing the citation, and instead has the placeholder "(REF)", which is unacceptable.

2. Data presentation is incomplete and inconsistent.
     a. Authors state they are analyzing 142 isolates, but numbers between ESBL (65) and non-ESBL (76) do not add up to 142. This is an issue throughout the text as well as in Table 1.
     b. Line 69 indicates 60 males, but Table 1 shows 61 males.
     c. Line 76-77: "MDR-E rate was significantly higher in women than men (69.7% vs. 30.3%)." Where are these numbers coming from? They do not coincide with Table 1.
     d. 20 males plus 43 females with ESBL-producing infections equals 63, not 65 as indicated at the top of Table 1. This is also an issue for non-ESBL-producing infections (41+37=78, not 76).
     e. In Table 1 and Table 3, use N(%) for consistency.
     f. In Table 1, 15-24 years, fix 19(13%) to use end parentheses instead of degree symbol.
     g. What is the logic behind the way in which the information is presented in Table 1 and Table 3? How is it organized, if at all?
     h. Table 1 title contains awkward phrasing. Consider revising.
     i. Figure 1 legend and caption include both English and French. Please keep consistent with a single language.
     j. Figure 1 should have a Y-axis title, such as "Percentage of Isolates Tested" or something similar.
     k. Not all of the paired columns in Figure 1 have their label visible. Please make the figure large enough to include all of the labels, not just every other one.
     l. Lines 82-88: the authors provide in the text the percentage of AMR in MDR-E and non-MDR-E groups, but Figure 1 appears to only present the MDR-E group. What is the rationale for this?
     m. Likewise, the numbers provided in the text do not match what is presented in the Figure (i.e. chloramphenicol; 7.7% in text vs. ~14% in Fig. 1). This needs remedied.
     n. Lines 113-116: the sequence types listed in the text do not perfectly align with those presented in Table 3 and vice versa. This is a result of either typographical errors or omission of data. This must be rectified.

Minor Concerns:
1. Italicize genus species names in Abstract, Lines 20-21.

2. Line 22, change "b" to either 'beta' or the Greek symbol b.

3. Lines 14-15 in Abstract do not coincide with Lines 36-37 in Introduction. Are UTIs the most frequent infections or the second most frequent infections? Needs correcting.

4. Resize columns in Table 2 to avoid word breaks (e.g. Carbapenemases and Resistance to quinolones).

5. Lines 136-138 contains a sentence with awkward phrasing.

6. Line 159: Cameroon is misspelled.

7. Line 190: ESBL-producing is missing the "L".

8. Line 223: sentences end/begin as "...EnterobacteriaceaeE. colistin MIC..." This needs fixed.

Author Response

  • “Abstract”
  1. Italicize genus species names in Abstract, Lines 20-21.

Answer: The correction was made

  1. Line 22, change "b" to either 'beta' or the Greek symbol b.

Answer: The correction was made

  • “Introduction”
  1. Lines 14-15 in Abstract do not coincide with Lines 36-37 in Introduction. Are UTIs the most frequent infections or the second most frequent infections? Needs correcting.

Answer: The correction was made

  • Results”

Comment N°1

  1. a. Please verify throughout the manuscript text which citation format you want to use. It's either text [1] or text[1] with or without the space between the text and the bracket. Select one and stick with it.

Answer: This is the text[1] without the space between the text and the parenthesis.

  1. Also verify proper spacing at the end of sentences that have citations. See Lines 37, 42, 44, 49, 52, 53, 129, 159, 188, 237, 247.

Answer: The correction was made

Comment N°2. Data presentation is incomplete and inconsistent.

  1. Authors state they are analyzing 142 isolates, but numbers between ESBL (65) and non-ESBL (76) do not add up to 142. This is an issue throughout the text as well as in Table 1.

Answer: The correction was made

  1. Line 69 indicates 60 males, but Table 1 shows 61 males.

   Answer: The correction was made

  1. Line 76-77: "MDR-E rate was significantly higher in women than men (69.7% vs. 30.3%).

" Where are these numbers coming from? They do not coincide with Table 1.

Answer: These are typographical errors the correction has been made

  1. 20 males plus 43 females with ESBL-producing infections equals 63, not 65 as indicated at the top of Table 1. This is also an issue for non-ESBL-producing infections (41+37=78, not 76).

Answer: The correction was made

  1. In Table 1 and Table 3, use N(%) for consistency.

Answer: The correction was made

  1. In Table 1, 15-24 years, fix 19(13%) to use end parentheses instead of degree symbol.

Answer: The correction was made

  1. What is the logic behind the way in which the information is presented in Table 1 and Table 3? How is it organized, if at all?

Answer: In Table 1, we have reported the total age percentage of ESBL-producing and non-producing isolates, but in Table 3 we describe the phylogenetic grouping of ESBL-producing and non-producing isolates; The information presented in the two tables is complementary.

  1. Table 1 title contains awkward phrasing. Consider revising.

Answer: The correction was made

  1. Figure 1 legend and caption include both English and French. Please keep consistent with a single language.

Answer: The correction was made

  1. Figure 1 should have a Y-axis title, such as "Percentage of Isolates Tested" or something similar.

Answer: The correction was made

  1. Not all of the paired columns in Figure 1 have their label visible. Please make the figure large enough to include all of the labels, not just every other one.

Answer: The correction was made

  1. Lines 82-88: the authors provide in the text the percentage of AMR in MDR-E and non-MDR-E groups, but Figure 1 appears to only present the MDR-E group. What is the rationale for this?

Answer: The correction was made

  1. Likewise, the numbers provided in the text do not match what is presented in the Figure (i.e. chloramphenicol; 7.7% in text vs. ~14% in Fig. 1). This needs remedied.

Answer: The correction was made

  1. Lines 113-116: the sequence types listed in the text do not perfectly align with those presented in Table 3 and vice versa. This is a result of either typographical errors or omission of data. This must be rectified.

Answer: The correction was made

Comment N°4. Resize columns in Table 2 to avoid word breaks (e.g. Carbapenemases and Resistance to quinolones).

Answer: The correction was made

  • Discussion”

Comment N°5. Les lignes 136-138 contiennent une phrase au phrasé maladroit.

Answer: The correction was made

Comment N°6. Line 159: Cameroon is misspelled.

Answer: The correction was made

Comment N°7. Line 190: ESBL-producing is missing the "L"

Answer: The correction was made

  • “Materials and Methods”

Comment N°8. Line 223: sentences end/begin as "...EnterobacteriaceaeE. colistin MIC..." This needs fixed.

Answer: The correction was made

Reviewer 3 Report

The manuscript of H.S. Mohamed titled "Multidrug-resistant Enterobacteriaceae in community-acquired urinary tract infections in Djibouti, Republic of Djibouti" is devoted to MDR Enterobacteriaceae in UTIs in Djibouti. Since the manuscript text was written by non-English-speaking authors, I would highly recommend English editing.

Some comments are below:

Line 2 and further in the manuscript text: please, carefully check that all Latin names (e.g. Enterobacteriaceae e.t.c.) and genes' names must be italicized.

Line 23: I suggest adding "Sanger" to "sequencing" to make it clear.

Figure 1: Replace "Sensible" with "Susceptible" and "Resistance" with "Resistant". Omit "de".

Table 1: Omit "-" between ST and 43. Place Klebsiella after all E. coli

Table 3: Replace "Molecular characterization" with "Sequence type". Omit C in EC

Line 159: Typo: Cameron -> Cameroon

Line 211: Add the manufacturer of Mueller-Hinton agar.

Line 223: Typo: Replace "EnterobacteriaceaE. colistin" with Enterobacteriaceae colistin"

Line 236: Replace (12-13) with [12-13]

Line 246: Omit "new"

Author Response

“Results”

Comment Table 1: Omit "-" between ST and 43. Place Klebsiella after all E. coli

Answer: The correction was made

Comment Line 159: Typo: Cameron -> Cameroon

Answer: The correction was made

Comment Line 211: Add the manufacturer of Mueller-Hinton agar.

Answer: The correction was made

Comment Line 223: Typo: Replace "EnterobacteriaceaE. colistin" with Enterobacteriaceae colistin"
Answer: The correction was made

Comment Line 236: Replace (12-13) with [12-13]
Answer: The correction was made

Comment Line 246: Omit "new"

Answer: The correction was made

Round 2

Reviewer 2 Report

In the revised manuscript, it appears that the authors have made substantive corrections to the manuscript. However, a few minor corrections remain that pertain to linguistic issues.

1) Line 23: replace sequencage (Fr) with sequencing (En)

2) Figure 1: replace sensible (Fr) with sensitive (En) in both graphs

3) Figure 1: replace resistance (Fr) with resistant (En) in the top graph

4) Line 93: replace with "Antibiotic resistance profile of Enterobacteriaceae isolates."

5) Line 142: return to original text.

6) Lines 226-227: replace with "...carbapenemase-producing E. coli. Colistin MIC was determined..."

7) Line 240: replace with "...PMQR-positive strains [53,54] were used as positive controls."

Author Response

Multidrug-resistant Enterobacteriaceae in community-acquired urinary tract infections in Djibouti, Republic of Djibouti

Hasna Said Mohamed , Mohamed Houmed Aboubaker , Yann Dumont , Marie-Noëlle Didelot , Anne-Laure Michon , Lokman Galal , Hélène Jean-Pierre , Sylvain Godreuil

Dear Editor in Chief,

Thank you for the attention you have given to this manuscript. We greatly appreciated the valuable comments and suggestions by the reviewers that helped us to significantly improve the manuscript

Reviewer #1

  • “Abstract”
  • Line 23: replace sequencage (Fr) with sequencing (En)

Answer: The correction was made.

  • Results”

2) Figure 1: replace sensible (Fr) with sensitive (En) in both graphs

Answer: The correction was made.

3) Figure 1: replace resistance (Fr) with resistant (En) in the top graph

Answer: The correction was made.

4) Line 93: replace with "Antibiotic resistance profile of Enterobacteriaceae isolates."

Answer: The correction was made.

5) Line 142: return to original text.

  Answer: The correction was made.

  • “Materials and methods”

6) Lines 226-227: replace with "...carbapenemase-producing E. coli. Colistin MIC was determined..."

    Answer: The correction was made.

7) Line 240: replace with "...PMQR-positive strains [53,54] were used as positive controls."

    Answer: The correction was made.